# ILRe: Intermediate Layer Retrieval for Context Compression in Causal Language Models

## Abstract

Large Language Models (LLMs) have demonstrated success across many benchmarks. However, they still exhibit limitations in long-context scenarios, primarily due to their short effective context length, quadratic computational complexity, and high memory overhead when processing lengthy inputs. To mitigate these issues, we introduce a novel context compression pipeline, called Intermediate Layer Retrieval (ILRe), which determines one intermediate decoder layer offline, encodes context by streaming chunked prefill only up to that layer, and recalls tokens by the attention scores between the input query and full key cache in that specified layer. In particular, we propose a multi-pooling combinations allocating strategy in the token retrieval process to maintain the completeness of semantics. Our approach not only reduces the prefilling complexity from $O(L^2)$ to $O(L)$ and trims the memory footprint to a few tenths of that required for the full context, but also delivers performance comparable to or superior to the full-context setup in long-context scenarios. Without additional post training or operator development, ILRe can process a single $1M$ tokens request in less than half a minute (speedup $\approx 180\times$) and scores RULER-$1M$ benchmark of $\approx 79.8$ with model Llama-3.1-UltraLong-8B-1M-Instruct on a Huawei Ascend 910B NPU.

## 1 Introduction

Large Language Models (LLMs) have achieved remarkable success across a wide range of context processing tasks, including document retrieval (Laban et al., 2023), code generation (Gu, 2023), multimodal applications (Sapkota et al., 2025), and agent systems (Guo et al., 2024; Luo et al., 2025). Especially in the long context scenarios, many works (Liu et al., 2025) are contributed to explore the limits of abilities of LLMs. Meanwhile, mainstream LLMs are increasingly supporting longer context lengths (Yang et al., 2025b; Xu et al., 2025; MiniMax et al., 2025).

Table 1: RULER benchmark performance of different methods with Llama-3.1-UltraLong-8B-1M-Instruct. Except the FullContext, the token budget is set to be 4K. For ILRe, Dual Chunk Attention (DCA) is optional. The best scores are in bold faces.

| Method | 32K | 64K | 128K | 512K | 1M |
|---|---|---|---|---|---|
| FullContext | 83.2 | 76.7 | 70.2 | OOM | OOM |
| StreamingLLM | 17.7 | 13.5 | OOM | OOM | OOM |
| SnapKV | 72.4 | 65.4 | OOM | OOM | OOM |
| RAG | 69.7 | 67.3 | 57.9 | 48.1 | 48.4 |
| ILRe w/ $l_R = 11$ (Ours) | 84.5 | 82.7 | 83.1 | 83.8 | 80.2 |
| ILRe w/ $l_R = 3$ (Ours) | **88.3** | **84.2** | **84.0** | 84.1 | 79.8 |
| ILRe w/ $l_R = 3$, DCA (Ours) | 86.8 | 83.5 | 83.5 | **84.3** | **84.0** |

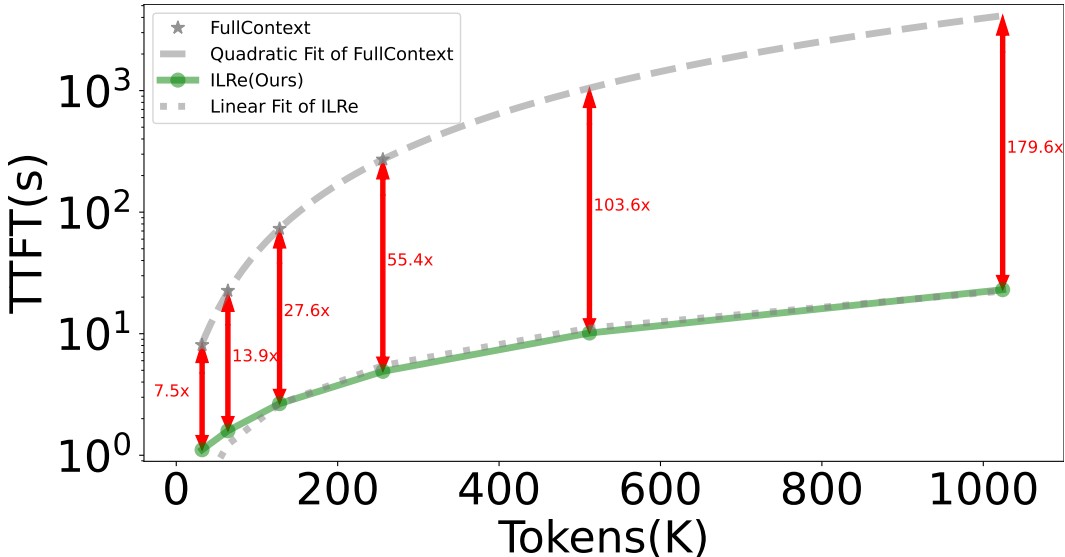

Figure 1: Time To First Token(TTFT) of Intermediate Layer Retrieval (ILRe) with Llama-3.1-8B, $l_R = 3$ and streaming chunked prefill on a Huawei Ascend 910B NPU. The red text notates the speedup of our approach.

However, LLMs face significant limitations in long-context applications. First, their effective context lengths are often much shorter than their claimed capacities. For instance, as shown in Table 1, a model claiming to support $1M$ tokens achieves $83.2$ accuracy at $32K$ tokens but drops sharply to $70.2$ at $128K$ in RULER benchmarks (Hsieh et al., 2024). Second, the quadratic computational complexity of attention leads to impractical prefilling times—reaching minutes for long contexts, as demonstrated in Figure 1. Finally, the memory overhead of key-value (KV) caches becomes prohibitive: for example, a $1M$-token cache with Llama-3.1-8B (Grattafiori et al., 2024) requires $\approx 120GB$ of memory, far exceeding the capacity of one typical GPU/NPU, even though the model already adapts GQA (Ainslie et al., 2023) technique to reduce the KV cache size.

Recently, many works have been contributed to mitigate the above issues. SnapKV (Li et al., 2024b) and Quest (Tang et al., 2024) proposed using the attention scores of an observation window to select salient tokens from the KV cache to reduce the complexity of decoding phase. DuoAttention (Xiao et al., 2025) introduced a streaming chunked prefill to encode incoming tokens within streaming heads in the prefilling phase. However, integrating these techniques into a unified, highly accurate, and efficient pipeline remains an open challenge. Meanwhile, our empirical observations reveal that different decoder layers exhibit varying accuracy of deserved content retrieved by SnapKV-like operation in which we use maximums of attention weights to unify focused content and combinations of max-pooling and average-pooling in the aggregation (see details in section 3.3). As illustrated in the "One Combination" series in Figure 2(a), some intermediate layers have high retrieval accuracy which also exhibit degradation when the length of key increases (same series in Figure 2(b)).

Inspired by the works and observations mentioned above, we propose a novel context compression method, **I**ntermediate **L**ayer **Re**treival (ILRe), as depicted in Figure 3. Our method processes the long full context with partial activated model parameters and streaming chunked prefill to reduce the prefilling complexity. In the layer determined by a simple retrieval task, it collects all the past key states of each chunk and the query states of question so that it can get full attention weights for query-aware global analysis. Specifically, we propose a strategy that evenly gathers retrieval tokens from multiple combinations of max- and avg-pooling to enhance the capability of retrieving long contents ("Multi Combinations" series in Figure 2).

We evaluate our approach with two benchmarks, RULER and LongBench (Bai et al., 2023), to demonstrate its effectiveness. The results show that our approach achieves remarkable performance across all benchmarks over other compression methods.

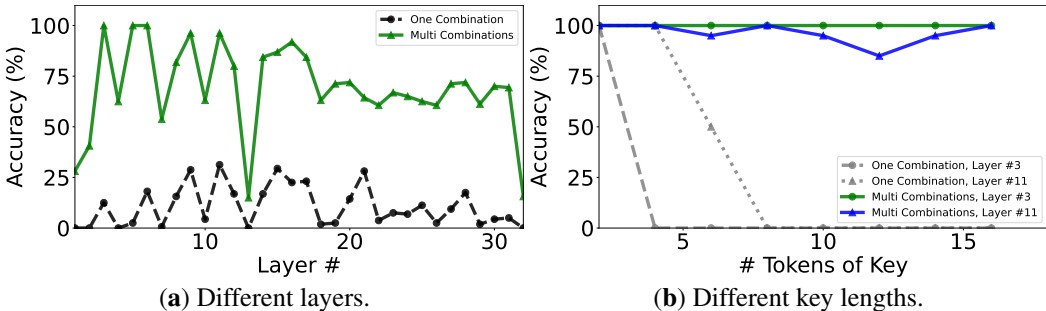

(a) Different layers.

(b) Different key lengths.

Figure 2: Retrieval Task results of model Llama-3.1-UltraLong-8B-1M-Instruct over 128K context. The context is prefilled by streaming chunked prefill. "One Combination" series is using one combination of max pooling with kernel size 1 and average-pooling with kernel size 5. "Multi Combination" series is using the default 48 combinations.

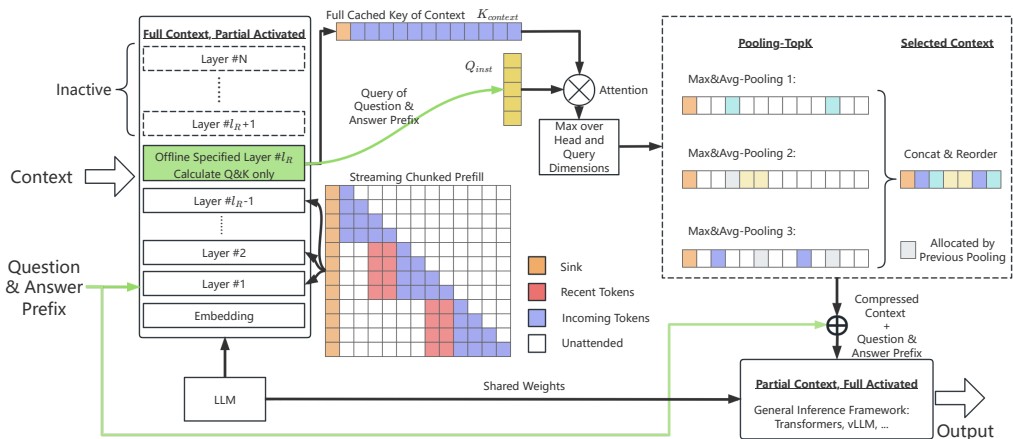

Figure 3: An Overview of Intermediate Layer Retreival(ILRe) framework.

## 2 RELATED WORKS

The limitations of large language models in long-context scenarios have inspired a range of approaches that aim to reduce memory usage, lower computation complexity, and preserve relevant information across extended sequences. Prior efforts broadly fall into the following two categories: internal memory compression and prompt-level context reduction.

### 2.1 INTERNAL MEMORY COMPRESSION

Autoregressive LLMs store key–value (KV) pairs during decoding to avoid recomputation, but this cache grows linearly with sequence length, creating a significant memory bottleneck. Several approaches have been proposed to mitigate this issue by selectively retaining or compressing the stored representations. H2O (Zhang et al., 2024) preserves only "heavy hitter" tokens that dominate attention scores, while FastGen (Ge et al., 2024) applies runtime profiling to dynamically choose compression policies. SnapKV (Li et al., 2024c) pools attention scores across prompt tokens to identify and retain informative clusters. Other works identify attention heads specialized for long-range recall and introduce head-specific cache compression (Wu et al., 2024; Fu et al., 2024b; Liu et al., 2024a; Jiang et al., 2024a). Instead of compressing KV states, another line of research explores how to reuse them more effectively. CacheBlend, for example, integrates cached KV segments from earlier prompts into the current decoding process to save recomputation (Yao et al., 2025). Along this line, Block-attention (Ma et al., 2025) introduces a blockwise mechanism that divides retrieved

documents into discrete blocks. This design allows previously seen blocks to reuse their KV states, reducing inference overhead in long-context scenarios like retrieval-augmented generation (RAG).

Beyond KV cache optimization, other efforts compress or reuse representations stored in intermediate layers. For example, layer-skipping approaches (Liu et al., 2024b) maintain a smaller subset of intermediate states for recall, balancing compression and semantic completeness. These researches align with other studies that examine the language model internal workflows(Liu et al., 2019), which show that intermediate layers captures increasingly abstract language patterns - from basic grammar to deeper meaning. Such insights motivate designs that exploit the representational richness of intermediate layers while reducing the memory and computation cost of long-context inference.

## 2.2 PROMPT-LEVEL CONTEXT REDUCTION

Prompt compression methods aim to reduce input length before model inference. Semantic compression approaches such as AutoCompressor (Chevalier et al., 2023) and gist-token-based prefix tuning (Mu et al., 2024) learn condensed prompt representations that preserve task-relevant semantics. Meanwhile, token-deletion techniques work at the token level, using metrics such as mutual information (Church & Hanks, 1989) or perplexity (Jiang et al., 2023) to prune less informative tokens.

Beyond these, recent work has proposed reinforcement learning (RL)-based, LLM scoring-based, and LLM annotation-based paradigms for prompt compression. KiS (Laban et al., 2021) addresses the simplification of unsupervised text by balancing fluency, significance, and simplicity through reinforcement learning. Using a k-SCST algorithm, it generates multiple candidate simplifications, evaluates them with a composite reward, and promotes candidates that exceed the mean reward. Similarly, SCRL (Li et al., 2024a) frames sentence compression as a sequence labeling task, fine-tuning a pre-trained transformer with a policy gradient method to classify tokens as essential or non-essential while optimizing for fluency and faithfulness. The selective context (Fu et al., 2024a) represents an LLM scoring-based approach that computes self-information for lexical units using a base causal language model, pruning tokens with low informativeness to yield a concise yet informative prompt. LLMLingua (Jiang et al., 2023) adopts a coarse-to-fine compression pipeline with a budget controller, iterative token-level pruning, and instruction tuning to maintain semantic integrity under high compression ratios. LongLLMLingua (Jiang et al., 2024b) extends this approach for long-context scenarios, incorporating question-aware compression, document reordering to reduce position bias (Yu et al., 2025), and dynamic compression ratios. LLMLingua-2 (Pan et al., 2024) further generalizes the method by introducing a GPT-4-distilled extractive compression dataset and defining compression as a Transformer-based token classification task that uses the full bidirectional context. In addition, a recent study proposes a retrieval framework that uses intermediate representations to obtain external information, which is then incorporated into the prompt to improve the final generation, showing consistent gains in multiple open-source LLMs(Lin et al., 2025).

## 2.3 CONTEXT LENGTH EXTRAPOLATION

A key challenge in deploying long-context models is their ability to generalize beyond their training sequence lengths(Zhao et al., 2024). Recent work has explored various approaches to enable length extrapolation, including positional encoding such as RoPE scaling(Su et al., 2023), YaRN(Peng et al., 2024). Other methods focus on architectural changes like sparse attention patterns(Lou et al., 2024) to extend context capabilities. While these approaches address the fundamental challenge of length generalization, they often require significant computational overhead or model retraining. Rather than scaling positions, Dual Chunk Attention (DCA)(An et al., 2024a) partitions long sequences into manageable chunks where all relative distances remain within the original training bounds, enabling models like LLaMA2-70B to handle contexts exceeding 100K tokens.

## 3 METHODOLOGY

In this section, we formally introduce our context compression method, ILRe. In this work, we focus on the decoder-only LLMs.

## 3.1 Preliminaries

The LLM uses its tokenizer to convert the words to input IDs, which are indices of the embedding metrics $E \in \mathbb{R}^{V \times d}$ of LLMs. Let $\mathbf{x} = (x_1, x_2, ..., x_L)$ be the input IDs of length $L$. The goal of prompt compression is to generate a subset $\tilde{\mathbf{x}} = \{\tilde{x}_i\}_{i=1}^{\tilde{L}}$ of $\mathbf{x}$, where $\tilde{L} < L$ is the allocation budget.

Collect the embeddings to form a input matrix, $X_1 \in \mathbb{R}^{L \times d}$. This matrix is the initial tokens of the first layer of LLM and then each layer will output a matrix of the same shape as the input of the next layer. For each head $h \in \{1, 2, ..., H\}$ in layer $l \in \{1, 2, ..., N_L\}$ with head dimension $d_h$, we focus on the Query and Key states, which are converted from tokens by three linear transformation matrices $W_{h,l}^Q, W_{h,l}^K \in \mathbb{R}^{d \times d_h}$ separately:

$$Q_{h,l} = PE(X_l W_{h,l}^Q), K_{h,l} = PE(X_l W_{h,l}^K) \tag{1}$$

where $PE$ stands for the positional encoding. The attention is calculated as:

$$A_{h,l} = \text{softmax}(\frac{Q_{h,l} K_{h,l}^T}{\sqrt{d_h}} \bigodot M_{h,l}) \tag{2}$$

where $\{M_{h,l}\}$ are attention masks. In a causal model, $M_{h,l}$ is usually a lower triangular matrix. But with the streaming chunked prefilling (Xiao et al., 2025), $M_{h,l}$ will be a $\Lambda$-like mask as demonstrated in Figure 3.

Rotary Positional Encodings (RoPE)(Sun et al., 2023) is currently the standard choice for $PE$ in LLMs. It's formalized as:

$$PE(\mathbf{X}) = \mathbf{R}(m)\mathbf{X} \tag{3}$$

where $m$ is the position index and $\mathbf{R}(m)$ is a rotation matrix.

## 3.2 Retrieval Tasks

We define a simple retrieval task to study the retrieval capability of each layer in LLM by embedding a number of numerical passkey phrases with varying lengths in random positions within a long textual context.

For each retrieval sample, we generate a numerical passkey by producing a random number of the desired length and padding it with leading zeros if necessary to ensure the number of digits. We also pair the passkey with a key ID string by randomly selecting three elements from a predefined set of common words, such as *dog, cat, yellow, red*, etc., and appending a two-digit number, joining them with hyphens. For example, one instance with $KEY\_ID = blue - cup - red - 33$ and $KEY\_VALUE = 198398$ is like "`\nThe blue-cup-red-33 magic passkey is 198398.\n`". Once generated, the passkey phrase is inserted into the background filler - a variant of the Paul Graham Essays[1] - at a systematically chosen position: the context is divided into 20 equal segments and the key is placed at the start of each segment in turn. The query part is "`\n\n# What's the <`$KEY\_ID$`> magic passkey?\nThe <`$KEY\_ID$`> magic passkey is `". The number of tokens in $KEY\_VALUE$ is the length of the key in this work instead of character number.

Following the *Needle-in-a-Haystack* setup (Kamradt, 2023), the evaluation spans multiple experimental setups that vary in *key depth*, *key length*, and *retrieval layer/combination configuration*. The budget of the retrieval task is set to 1024. We evaluate performance using exact match accuracy, where a prediction is correct only if the entire passkey phrase is retrieved without error.

## 3.3 Retrieval Method

We use the attention between the context and the query to identify the critical tokens for a specific query $q$. The attention is calculated as:

$$A_{h,l}^{q,C} = \text{softmax}\left(\frac{q_{h,l} K_{h,l}^{C}{}^T}{\sqrt{d_h}}\right) \tag{4}$$

---

[1]`https://paulgraham.com/articles.html`

**Algorithm 1** Context Allocation

---

**Input**: attention weights $A^C$ without sink part, max pooling kernel sizes $\{m_i\}_{i=1}^{N_M}$, average pooling kernel sizes $\{n_j\}_{j=1}^{N_A}$, total budget $B$, input IDs $C$;

**Output**: compressed context $\tilde{C}$

    Init a set of allocated indices $I$ with indices of sink part
    Calculate the budget of each combination $B_c = \frac{B}{N_M N_A}$
    **for** $i \leftarrow 1$ to $N_M$ **do**
        Calculate the k of top in this max pooling $K_m = \frac{B}{m_i} + 1$
        Calculate $\tilde{A}_{m_i}^C = \text{MaxPooling}(A^C)$ with $size = m_i$, $stride = m_i$
        **for** $j \leftarrow 1$ to $N_A$ **do**
            Calculate $\tilde{A}_{m_i,n_j}^C = \text{AvgPooling}(\tilde{A}_{m_i}^C)$ with $size = n_j$, $stride = 1$
            Select top $K_m$ indices from $\tilde{A}_{m_i,n_j}^C$ as $\tilde{I}_{m_i,n_j}$
            Map $\tilde{I}_{m_i,n_j}$ back to indices of $C$ as $I_{m_i,n_j}$
            Let $S_I = 0$
            **while** $S_I < B_c$ **do**
                Get next index $I_{m_i,n_j}^k \leftarrow I_{m_i,n_j}$
                **if** $I_{m_i,n_j}^k \notin I$ **then**
                    Update $I = I \cup I_{m_i,n_j}^k$
                    Update $S_I = S_I + 1$
                **end if**
            **end while**
        **end for**
    **end for**
    Sort $I$ in ascending order
    Gather input IDs from $C$ by indices $I$ as new context $\tilde{C}$ **Return** compressed context $\tilde{C}$

---

Here, we assume that the batch size is 1, that is, $A_{h,l}^{q,C}$ is a 3D matrix in the layer $l$ with head, query, and context dimensions. Let $L_q$ denote the size of the query $q$; typically, $L_q$ is much smaller than the length $L$ of the context $C$. In the process of context compression, we need to determine which input IDs of context (rather than KV pairs) should be retained for inference. To achieve this, we must reduce the attention from 3D to a 1D matrix with only the context dimension left. For that, we only keep the maximums of attention across the head and query dimensions in one specific layer $l_R$:

$$A^C = \max_{h,q} A_{h,l_R}^{q,C} \tag{5}$$

Since we calculate the full attention in the layer $l_R$ only, we omit the symbol $l_R$ for convenience, and, in practice, only the key states will be cached with full length of $L$ in that layer.

Many works, like SnapKV, are using average pooling preserves semantic features rather than only retaining those with the highest attention weights. Here, we extend this technique by incorporating combinations of different pooling choices and budget allocation. This procedure is detailed in Algorithm 1. SnapKV constitutes a special case of this algorithm, featuring one max pooling with kernel size 1 and one average pooling of kernel size. By default, we use 3 max pooling with kernel sizes $2, 4, 8$ and 16 average pooling with kernel sizes from 1 to 16. This results in 48 combinations of max and average pooling. We evenly distribute the token budget across all combinations. If a token has already been allocated by previous combinations, the next-highest-ranked candidate is selected instead, continuing until the target budget is met.

In the simple retrieval task defined in section 3.2, we calculate the retrieval accuracy of each layer across varying key depths and lengths, as presented in Figure 2. We observe that certain layers equipped with only a single combination of max and average pooling can achieve a $100\%$ retrieval accuracy in short-key tasks but decline sharply as the key size increases. In contrast, our multi-combination approach maintains consistent performance across different key sizes. Furthermore, frontier layers exhibit higher recall rates than later layers, resulting in the optimal $l_R$ typically being smaller than half of $N_L$ and thus yielding much lower computational complexity. We define the optimal layer $l_R$ as the layer with the smallest index value among those exhibiting the highest recall

rates. For example, in the model Llama-3.1-Ultralong-8B-1M-Instruct (Xu et al., 2025), we identify the optimal $l_R = 3$ versus the total number of layers $N_L = 32$. Meanwhile, we count the frequency of occurrence of maxima arguments of attention heads in Appedix A.2, which demonstrates that attention heads are not fixed in their behavior.

## 3.4 INTERMEDIATE LAYER RETRIEVAL

The main schema of Intermediate Layer Retrieval(ILRe) is present in Figure 3. For end-to-end inference:

- Determine $l_R$ offline as discussed above. Load LLM parameters up to the layer $l_R$.
- Prepare a long context and short query part. Usually, the query part consists of a question and an answer prefix. Split the context $C$ into chunks $\{C_i\}_{i=1}^{N_C}$.
- In layers $\{1, 2, ..., l_R - 1\}$, apply the streaming chunked prefill. Only cache the sink and sliding part of the key states and value states with sizes $S$ and $W$ respectively.
- In the retrieval layer $l_R$, calculate the key states for the context, and then skip the rest of the calculations. Pre-allocate memory with context length $L$ for key states caching $K_{h,l_R}^C$ and fill the cache with chunked key states $\{K_{h,l_R}^{C_i}\}_{i=1}^{N_C}$.
- Prefill the query part to get the query states $q_{h,l_R}$.
- Calculate $A^C$ with the cached key states and the query states in the layer $l_R$.
- Apply algorithm 1 to select input IDs.
- Concatenate the selected input IDs with those of the query part to form a new input for standard inference. This component can be offloaded to other highly optimized inference frameworks, such as vLLM (Kwon et al., 2023), SGLang (Zheng et al., 2024), and others.

**Dual Chunk Attention (DCA)(An et al., 2024b).** As proposed in DCA, the relative positional encoding with the bound distance can effectively extrapolate the length capacity of LLMs. We adapt DCA to ILRe in the following way. For the sink parts $K_{l,h}^S$ in layers $\{1, 2, ..., l_R - 1\}$, we re-RoPE them by the chunk size as the distance $D_C$ in each chunk prefilling since the third chunk. That is

$$K_{l,h}^{S,i} = \mathbf{R}^{i-2}(D_C)K_{l,h}^S \text{ for } l \in \{1, 2, ..., l_R - 1\} \text{ and } i > 2 \text{ only.} \quad (6)$$

For the key states in the layer $l_R$, we re-RoPE them by the following equation before retrieving tokens:

$$\tilde{K}_{h,l_R}^{C_i} = \mathbf{R}(D_i)K_{h,l_R}^{C_i} \text{ for } i < N_C - 1 \text{ only.} \quad (7)$$

where $D_i$ is the distance between chunk $i$ and chunk $N_C - 1$.

We select the chunk $N_C - 1$ as the positional base in considering the last chunk possible shorter than others. So, the distance between all chunked key states and the query state will be bounded in the range of two times the chunk size.

## 3.5 COMPLEXITY ANALYSIS

We only discuss the attention complexity in the context compression, as the standard inference of a compressed prompt is constrained by the fixed budget $B$.

Table 2: Complexity of Computation and Memory footprint. For convenience, we set $W' = S + W$ and $l_R' = l_R - 1$. The factor 2 is for the key and value states.

| Method | Computation | Memory |
|---|---|---|
| FlashAttention | $O(N_L L^2)$ | $O(2N_L L)$ |
| ILRe(Ours) | $O((W' l_R' + L_q)L)$ | $O(2W' l_R' + L)$ |

The complexity of our approach is listed in Table 2. Compared to the standard FlashAttention (Dao, 2023), ILRe only calculates attentions in a streaming chunk fashion in $l_R - 1$ layers and in an

Table 3: Longbench results of model Llama-3.1-8B-Instruct and Qwen2.5-7B-Instruct. The best and second best scores in each group are boldfaced and underlined respectively.

| Model | B | Method | Single Doc. QA | | | Multi Doc. QA | | | Summarization | | | Few-shot Learning | | | Synthetic | | Code | | Avg. |
|---|---|---|---|---|---|---|---|---|---|---|---|---|---|---|---|---|---|---|---|
| | | | NarrativeQA | Qasper | MF-en | HotpotQA | 2WikiMQA | Musique | GovReport | QMSum | MultiNews | TREC | TriviaQA | SAMSum | PCount | PRe-en | Lcc | RB-P | |
| Llama-3.1-8B-Instruct | | FullContext | 23.5 | 45.4 | 54.3 | 44.8 | 41.6 | 24.1 | 34.4 | 23.6 | 27.4 | 69.5 | 91.2 | 44.4 | 6.6 | 76.0 | 63.2 | 54.9 | 45.3 |
| | 2K | SnapKV | **30.8** | 42.8 | 52.3 | 53.4 | **44.5** | 29.3 | 30.3 | **24.8** | 27.0 | 2.0 | **91.6** | 8.3 | 4.5 | **99.5** | 18.6 | 15.0 | 35.9 |
| | | StreamingLLM | 22.9 | 34.3 | 37.1 | 43.9 | 34.4 | 18.0 | 28.1 | 22.2 | 26.8 | 0.0 | 72.8 | 8.4 | 2.4 | 95.0 | 14.6 | 12.9 | 29.6 |
| | | RAG | 18.6 | 41.8 | 48.4 | 45.5 | 40.8 | 24.7 | 30.9 | 23.1 | 26.9 | 67.0 | 91.0 | 40.7 | 2.0 | 67.0 | 57.8 | 49.1 | 42.2 |
| | | ILRe w/ $l_R = 3$ | 22.0 | 42.0 | 54.6 | 53.1 | 34.7 | 26.8 | 30.7 | 22.3 | 27.1 | 71.0 | 91.2 | 37.9 | 4.8 | 72.5 | 63.9 | 55.6 | 44.4 |
| | | ILRe w/ $l_R = 6$ | 20.4 | 43.8 | 55.2 | 57.5 | 41.6 | 27.3 | 31.1 | 24.0 | 27.0 | 69.0 | 90.6 | 43.2 | 3.1 | 76.5 | 59.6 | 49.8 | 45.0 |
| | | ILRe w/ $l_R = 10$ | 22.0 | 43.5 | 50.9 | 50.7 | 39.2 | 26.7 | 31.4 | 20.6 | 27.2 | 68.0 | 89.6 | 37.8 | 3.9 | 83.5 | 63.6 | 56.3 | 44.7 |
| | | ILRe w/ $l_R = 11$ | 26.0 | 45.4 | 53.6 | 55.3 | 40.2 | 29.2 | 31.5 | 22.4 | 27.1 | 73.5 | 89.9 | 38.2 | 4.6 | 83.0 | 64.3 | 56.9 | 46.3 |
| | | ILRe w/ FullKV,$l_R = 11$ | 26.3 | 47.2 | 56.2 | 58.6 | 43.1 | 30.6 | 31.6 | 24.8 | 26.9 | 71.0 | 91.3 | 43.4 | 4.0 | 83.5 | 60.4 | 48.2 | 46.7 |
| | 1K | SnapKV | 29.4 | 40.9 | 51.8 | 53.1 | 45.7 | 29.2 | 27.6 | 24.0 | 26.0 | 0.5 | 91.4 | 11.3 | 4.0 | 99.5 | 18.3 | 16.1 | 35.6 |
| | | StreamingLLM | 22.4 | 27.0 | 35.6 | 42.2 | 26.1 | 15.4 | 25.6 | 21.6 | 25.9 | 0.0 | 70.5 | 9.4 | 2.8 | 97.0 | 11.6 | 15.2 | 28.0 |
| | | RAG | 20.5 | 35.4 | 45.4 | 46.7 | 40.6 | 22.9 | 28.0 | 22.0 | 25.9 | 62.0 | 87.4 | 38.0 | 2.1 | 46.0 | 53.3 | 48.1 | 39.0 |
| | | ILRe w/ $l_R = 3$ | 18.1 | 33.2 | 51.5 | 49.2 | 34.8 | 21.6 | 28.6 | 20.1 | 26.2 | 66.0 | 85.7 | 37.6 | 2.5 | 36.5 | 63.6 | 54.4 | 39.4 |
| | | ILRe w/ $l_R = 6$ | 19.0 | 40.2 | 53.7 | 56.8 | 36.0 | 23.4 | 29.0 | 22.8 | 25.8 | 66.5 | 89.0 | 41.6 | 2.9 | 45.5 | 54.7 | 50.1 | 41.1 |
| | | ILRe w/ $l_R = 10$ | 19.3 | 35.9 | 43.1 | 43.1 | 29.9 | 16.7 | 29.6 | 17.3 | 26.4 | 59.5 | 79.4 | 36.6 | 4.6 | 46.5 | 61.2 | 53.6 | 37.7 |
| | | ILRe w/ $l_R = 11$ | 23.4 | 44.2 | 50.2 | 50.9 | 37.0 | 23.7 | 29.2 | 19.5 | 26.0 | 70.0 | 85.2 | 36.4 | 2.6 | 56.0 | 62.3 | 54.3 | 41.9 |
| | | ILRe w/ FullKV,$l_R = 11$ | 24.1 | 44.1 | 54.3 | 56.9 | 42.1 | 25.6 | 29.6 | 22.6 | 25.8 | 64.5 | 88.6 | 43.4 | 3.2 | 42.5 | 54.7 | 52.1 | 42.1 |
| Qwen2.5-7B-Instruct | | FullContext | 29.3 | 44.2 | 52.6 | 58.0 | 46.9 | 30.5 | 31.7 | 23.4 | 24.0 | 72.5 | 89.3 | 45.8 | 8.0 | 100.0 | 60.6 | 66.9 | 49.0 |
| | 2K | SnapKV | 28.4 | 44.5 | 51.0 | 55.1 | 42.1 | 32.7 | 30.0 | 23.5 | 25.1 | 60.0 | 88.9 | 40.6 | 10.5 | 100.0 | 5.2 | 7.1 | 40.3 |
| | | StreamingLLM | 23.2 | 37.7 | 33.2 | 41.6 | 35.1 | 19.6 | 28.5 | 20.8 | 25.0 | 31.0 | 71.0 | 38.6 | 9.5 | 34.5 | 12.8 | 14.8 | 29.8 |
| | | RAG | 19.8 | 41.7 | 47.7 | 50.9 | 41.1 | 24.8 | 28.8 | 22.3 | 23.5 | 68.0 | 78.7 | 42.4 | 3.0 | 61.5 | 53.9 | 42.1 | 40.6 |
| | | ILRe w/ $l_R = 5$ | 18.6 | 38.3 | 52.1 | 54.7 | 41.7 | 30.0 | 29.1 | 22.9 | 23.8 | 68.5 | 89.0 | 42.8 | 3.5 | 60.5 | 61.2 | 62.1 | 43.7 |
| | | ILRe w/ $l_R = 6$ | 19.1 | 40.3 | 44.3 | 51.4 | 41.6 | 27.6 | 29.6 | 21.6 | 23.6 | 67.5 | 89.2 | 43.8 | 6.0 | 60.0 | 59.9 | 62.6 | 43.0 |
| | | ILRe w/ $l_R = 9$ | 20.5 | 40.0 | 50.5 | 56.7 | 45.7 | 32.6 | 29.9 | 21.8 | 23.6 | 69.5 | 90.5 | 43.4 | 6.0 | 69.0 | 59.5 | 62.2 | 45.1 |
| | | ILRe w/ FullKV,$l_R = 9$ | 21.5 | 43.4 | 51.3 | 58.9 | 46.6 | 31.0 | 29.4 | 22.2 | 23.6 | 69.0 | 89.8 | 41.9 | 6.0 | 75.5 | 59.3 | 61.8 | 45.7 |
| | 1K | SnapKV | 27.9 | 42.3 | 49.4 | 52.7 | 41.4 | 29.7 | 27.4 | 23.1 | 24.0 | 58.5 | 88.7 | 40.4 | 10.5 | 100.0 | 8.0 | 7.7 | 39.5 |
| | | StreamingLLM | 21.4 | 30.0 | 28.4 | 38.6 | 32.3 | 15.4 | 26.5 | 20.1 | 24.5 | 30.0 | 57.6 | 38.9 | 9.5 | 25.0 | 16.0 | 16.4 | 26.9 |
| | | RAG | 17.3 | 34.0 | 44.0 | 45.2 | 37.3 | 25.4 | 27.1 | 22.1 | 22.9 | 61.5 | 62.0 | 39.2 | 5.0 | 45.0 | 45.5 | 39.2 | 35.8 |
| | | ILRe w/ $l_R = 5$ | 17.0 | 32.5 | 49.4 | 53.4 | 35.3 | 24.2 | 27.9 | 21.0 | 23.1 | 62.5 | 83.3 | 43.1 | 4.0 | 38.5 | 61.8 | 59.9 | 39.8 |
| | | ILRe w/ $l_R = 6$ | 13.6 | 31.1 | 39.0 | 47.4 | 39.1 | 19.9 | 28.0 | 19.9 | 23.2 | 63.5 | 88.5 | 42.5 | 5.5 | 36.0 | 61.4 | 58.4 | 38.6 |
| | | ILRe w/ $l_R = 9$ | 18.1 | 33.8 | 49.3 | 51.6 | 37.2 | 26.4 | 28.9 | 20.6 | 23.1 | 67.5 | 85.9 | 43.0 | 5.0 | 40.5 | 60.2 | 57.3 | 40.5 |
| | | ILRe w/ FullKV,$l_R = 9$ | 17.6 | 37.9 | 51.5 | 57.3 | 39.3 | 27.8 | 28.8 | 20.9 | 23.2 | 66.5 | 85.7 | 41.7 | 4.5 | 47.5 | 61.5 | 58.3 | 41.9 |

observation window manner in the specific layer $l_R$. As $S, W, L_q \ll L$ in the long context scenarios, the computation complexity of ILRe approximates $O(L)$; notably, the memory footprint is roughly $\frac{1}{2N_L}$ of that of the full context.

## 4 EXPERIMENTS

In this section, we empirically evaluate the effectiveness of our approach, ILRe.

**Baseline Methods.** For comparison, except the original prompt with full context, we select other methods similar to or used in the components of our approaches: StreamingLLM (Xiao et al., 2023), SnapKV (Li et al., 2024b), and Retrieval-Augmented Generation (RAG) (Gao et al., 2024; Gupta et al., 2024) with Qwen3-Embedding-0.6B (Zhang et al., 2025) as the backbone. RAG uses 256 tokens as the text chunk size.

**Benchmarks and LLMs.** In this work, we use two benchmarks: LongBench and RULER. For details of setups in these benchmarks, see Appendix A.3.

**Default Settings of ILRe.** By default, we set $S = 4$, $W = 512$, and the chunk size of prefilling is 1024(the first chunk with additional $S$). For the pooling, we use 3 max pooling with kernel sizes 2, 4, 8 and 16 average pooling with kernel sizes from 1 to 16 as mentioned in Section 3.3. DCA is off unless it is specified.

**Specific Terms.** The term "FullKV" refers to $W = \infty$ and "FullContext" for whole-context inference with full KV cache retention.

### 4.1 RULER

The RULER results are presented in Table 1. Our approach consistently outperforms other methods, including the full context with original prompt. Furthermore, performance degradation exhibits a much weaker dependence on context length compared to other methods.

## 4.2 LONGBENCH

We evaluate our method on the LongBench benchmark, which measures long-context understanding across a diverse set of tasks, including question answering, summarization, fewshot learning, synthetic tasks, and coding. Table 3 reports the results under token budget of 2K and 1K.

We highlight two key observations. First, compared with other compression algorithms, ILRe demonstrates competitive performance at 2K token budget. ILRe with full KV retention attains the best overall performance, outperforming SnapKV and StreamingLLM in most tasks, with particularly strong gains in retrieval-heavy tasks such as QA and summarization. Even with various $l_R$, ILRe remains competitive, demonstrating that intermediate layer retrieval can recover useful long-context information with fewer stored tokens. Second, at 1K context length, ILRe continues to scale favorably, showing robustness under tighter memory budgets.

## 4.3 ABLATIONS

Table 4: Different prefilling methods with Llama-3.1-UltraLong-8B-1M-Instruct in RULER.

| Settings | 32K | 64K | 128K |
|---|---|---|---|
| Streaming Chunked Prefill | 88.3 | 84.2 | 84.0 |
| FullKV Prefill | 87.2 | 83.5 | 84.0 |

**Prefilling Method.** To demonstrate the stability of the streaming chunked prefill in our framework, we compare it with using the full KV in layers $\{1, 2, ..., l_R - 1\}$ by setting $W = \infty$. As shown in Table 4, the results of RULER benchmark with the streaming chunked prefill are comparable to those of full KV. The results with "FullKV" in Table 3 further reinforce this finding.

Table 5: RULER benchmark performance of different pooling settings with Llama-3.1-UltraLong-8B-1M-Instruct. The best average scores are in bold faces.

| Settings | 128K | | | 512K | | |
|---|---|---|---|---|---|---|
| | NS2 | NS3 | Avg. | NS2 | NS3 | Avg. |
| Default | 100.0 | 100.0 | **84.0** | 100.0 | 100.0 | **84.1** |
| Less-Max-Combinations | 100.0 | 99.2 | 82.4 | 100.0 | 99.4 | 82.8 |
| Less-Avg-Combinations | 100.0 | 100.0 | 82.5 | 100.0 | 100.0 | 83.1 |
| One-Max-Avg | 100.0 | 0.0 | 65.6 | 100.0 | 1.2 | 66.6 |

**Multi Combinations.** We define three different configurations of the pooling method from the default setups. **Less-Max-Combinations** uses only one max pooling with kernel size 4. **Less-Avg-Combinations** reduces the number of average pooling to nine with sizes ranging from 1 to 9. **One-Max-Avg** only has one max pooling and one average pooling with size 4 and 5 respectively.

These tasks are evaluated at $128K$ and $512K$ in the RULER benchmarks present in Table 5. We include two subtasks, NIAH-Single-Key-2 (NS2) and NIAH-Single-Key-3 (NS3), to demonstrate that multi-combination are effective for long-content retrieval. This is evident as the One-Max-Avg setup exhibits an abrupt performance drop in the longer-key task NS3.

**Length Extrapolation.** We evaluate LLMs without lengthy tuning using the RULER benchmark (see Table 6), aiming to verify the compatibility of DCA with ILRe.

The default ILRe (without any extrapolation technique) performs well under $128K$ tokens across all tested models—even for the Qwen family with a native context window capacity of only $32K$ tokens. When DCA is enabled, ILRe further extends the effective context length of LLMs. Notably,

Table 6: ILRe with/without Dual Chunk Attention in RULER benchmark. Because the long context measurements are time consuming, we halt the measurements after the score is below 30. Qwen3-32B (Yang et al., 2025a) is measured only in the non-thinking mode.

| Model | Claimed Length | Method | 4K | 8K | 16K | 32K | 64K | 128K | 256K | 512K | 1M |
|---|---|---|---|---|---|---|---|---|---|---|---|
| Llama3.1-8B-Instruct | 128K | FullContext | 95.7 | 93.9 | 91.4 | 86.6 | 83.8 | 74.8 | OOM | OOM | OOM |
| | | ILRe w/ $l_R = 3$ | - | 93.5 | 90.6 | 87.1 | 85.3 | 76.8 | 45.6 | 17.2 | - |
| | | ILRe w/ $l_R = 3$, DCA | - | 92.8 | 90.1 | 87.8 | 84.8 | 85.6 | 81.9 | 83.7 | 83.2 |
| Qwen2.5-7B-Instruct | 32K | FullContext | 94.5 | 92.6 | 91.8 | 89.0 | 69.4 | 25.8 | OOM | OOM | OOM |
| | | ILRe w/ $l_R = 5$ | - | 92.4 | 91.9 | 89.1 | 84.8 | 83.2 | 53.8 | 13.1 | - |
| | | ILRe w/ $l_R = 5$, DCA | - | 92.6 | 92.2 | 89.5 | 85.6 | 84.7 | 83.8 | 84.7 | 83.5 |
| Qwen3-32B | 32K | FullContext | 95.5 | 94.8 | 94.8 | 93.7 | OOM | OOM | OOM | OOM | OOM |
| | | ILRe w/ $l_R = 6$ | - | 94.7 | 93.0 | 89.7 | 86.7 | 85.0 | 20.4 | - | - |
| | | ILRe w/ $l_R = 6$, DCA | - | 95.4 | 94.0 | 89.6 | 87.0 | 86.7 | 85.1 | 84.2 | 20.7 |

results for the Llama family (presented in Table 1 and 6) demonstrate that Llama3.1-8B-Instruct, with both ILRe and DCA enabled, achieves results comparable to Llama-3.1-UltraLong-8B-1M-Instruct across the $128K$–$1M$ token range.

## 4.4 EFFICIENCY

In Figure 1, we measure the Time To First Token (TTFT) of our approach with a batch size of one on a single Huawei Ascend 910B NPU, averaging results across multiple trials. Due to hardware limitations, we only measure the FullContext baseline up to $256K$ tokens, and extend the results to $1M$ tokens using a quadratic fit. Given the large discrepancy in TTFT between our approach and the full-context baseline, we use the log-scale y-axis in the figure. Our method demonstrates a significant speedup, achieving up to a hundredfold improvement in long-context inference. Additionally, the linear fit of ILRe yields an $R^2$ value of $0.994$, which strongly aligns with our complexity analysis conclusion of $O(L)$ in Section 3.5.

## 5 CONCLUSION

In this study, we propose a training-free context compression framework, ILRe. Built upon decoder-only causal LLMs, it is highly efficient and effective in context prefilling and extraction. By integrating streaming chunked prefill, an observation window, multi pooling combinations and an early exiting layer, ILRe reduces the computational overhead of self-attention, delivering up to a hundred-fold speedup in end-to-end inference while maintaining remarkable accuracy. Our experimental evaluations demonstrate that ILRe holds significant promise for advancing long-context LLM inference and length extrapolation.

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

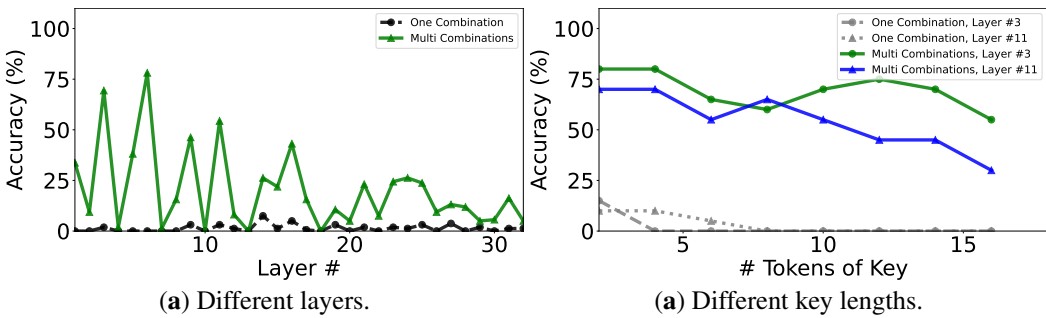

(a) Different layers.   (a) Different key lengths.

Figure 4: Retrieval Task results of model Llama-3.1-8B-Instruct over $128K$ context.

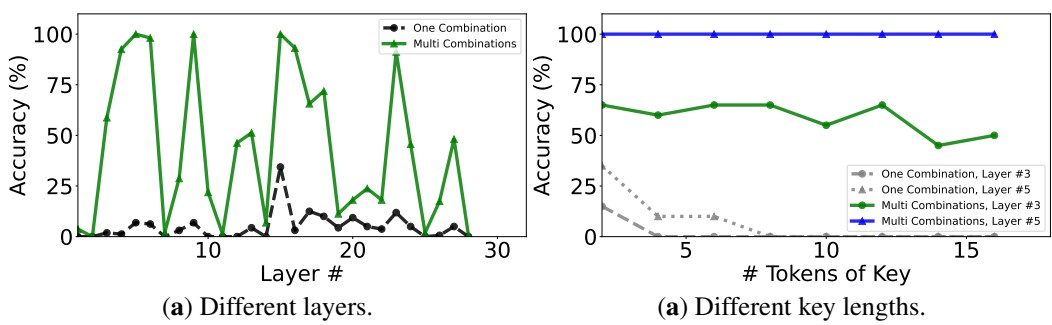

(a) Different layers.   (a) Different key lengths.

Figure 5: Retrieval Task results of model Qwen2.5-7B-Instruct over $32K$ context.

## A APPENDIX

### A.1 MORE RESULTS OF RETRIEVAL TASK

Similar to Figure 2, the results of Llama-3.1-8B-Instruct and Qwen2.5-7B-Instruct in the Retrieval Task are present in Figure 4 and 5 respectively.

### A.2 OCCURRENCE OF MAX ATTENTION HEADS

In Figure 6, the statistics of the maximum attention head indices in the retrieval task are present. This data is aggregated after the max operation of attention matrix over the query dimension.

### A.3 SETUPS OF BENCHMARKS

For RULER, we directly use the codebase of (Hsieh et al., 2024) and fix the to-ken budget to $4K$. We extract the question and answer prefix from the input prompt as the query part(with the remaining tokens serving as context). For example, the query for Qwen2.5-7B-Instruct is like "`Question: In what country is Normandy located?\n<|im_end|>\n<|im_start|>assistant\n\n Answer:`".

For LongBench, we utilize the toolset introduced in (Yuan et al., 2024), and extract the last 64 tokens from each input prompt as the query (also with the remaining tokens serving as context). Given that LongBench contexts are relatively short (with an average length of fewer than $16K$ tokens), we only test it with two short length capacity models: Llama-3.1-8B-Instruct (Grattafiori et al., 2024) and Qwen2.5-7B-Instruct (Qwen et al., 2025).

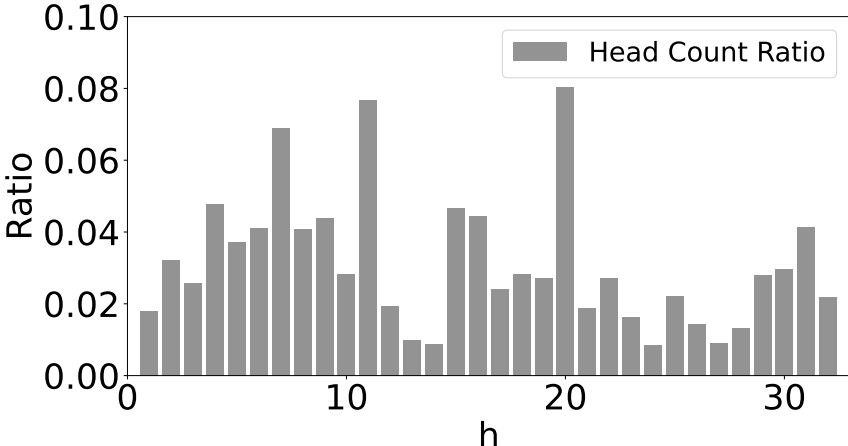

Figure 6: Occurrence Ratios of max attention head indices in one $128K$ Retrieval Task of model Llama-3.1-Ultralong-8B-1M-Instruct.

