# OpenReview forum: "ILRe: Intermediate Layer Retrieval for Context Compression in Causal Language Models"
_ICLR.cc/2026/Conference — Submitted to ICLR 2026_

### Official Review · Reviewer_Hihf · 2025-10-25

**Soundness:** 2
**Presentation:** 1
**Contribution:** 2
**Rating:** 2
**Confidence:** 5

**Summary:**

The paper introduces ILRe (Intermediate Layer Retrieval), a training‑free context‑compression pipeline for decoder‑only LLMs. Specifically, ILRe: (i) select an intermediate decoder layer $l_R$ offline using a retrieval probe, (ii) during inference, streamingly prefill the long context with attention sinks and sliding windows only up to $l_R$ , cache full keys at that single layer, and (iii) compute query–context attention at $l_R$ and select tokens for a compressed prompt via a multi‑kernel pooling scheme. Across RULER and LongBench, ILRe is competitive with or better than full‑context inference at lower cost, outperforming all compared baselines.

**Strengths:**

1. **Careful ablation studies.** The settings of analysis experiments are appropriate and clear.

**Weaknesses:**

1. **Unclear presentation.** Some critical definitions in this article are hard to understand. In the Introduction, the author proposed the concept of "kernels", and gave Figure 2, illustrating the performance of different kernels. However, (i) "kernels" is not well-defined after proposing, and (ii) the explanations to results in figure 2 (Line 84-86) are confusing, as I have no idea how "intermediate layers" is presented in the figures, and how is "degradation when the length of key increases" presented by the figures. Moreover, the methodology section is also very confusing. Section 3.2 describes observations of retrieval tasks without introducing what is the retrieval task and how to perform the task. Inside Section 3.2, KVCache-related compression descriptions occupies most of the content, and is very misleading upon first-time reading. Furthermore, Figure 3 seems to illustrate the method differently from the article, as I couldn't match each component in Algorithm 1, Section 3 to Figure 3. Overall, the entire article needs to be re-written and re-organized for better presentation.
2. **Lack of comparisons with the latest baselines.** A number of notable baselines are not compared, and the latest compared methods are published nearly 2 years ago. For instance, [1-3] are some latest and representative works that should be noticed. The authors are recommended to add at least 2 baselines published at (or after) early 2025 (not limited to [1-3]).
3. **Limited experiment benchmarks.** More comparing metrics, such as PPL, and on reasoning tasks (i.e., L-Eval[4]), are recommended.
4. **Novelty issues.** ILRe seems like an extension of SnapKV with more delicate kernel and attention mask choices, which limits the novelty of the proposed method.
5. **Formatting.** This article **exceeds the 9-page limit**.

[1] LongLoRA: Efficient Fine-tuning of Long-Context Large Language Models

[2] LLM Maybe LongLM: Self-Extend LLM Context Window Without Tuning

[3] Extending LLM Context Window with Adaptive Grouped Positional Encoding: A Training-Free Method

[4] L-Eval: Instituting Standardized Evaluation for Long Context Language Models

**Questions:**

See weaknesses.

---

> ### Author Response · Authors · 2025-11-14
>
> For weaknesses 1,4,5: please refer to the revision comment.
>
> For weakness 2: In this work, we propose a compression method. However ref [1-3] are methods on tuning or extrapolations. We can put them in the related work Sec. 2.3 in the final revision.
>
> For weakness 3: PPL is demonstrated that it's inappropriate in the long context scenarios by the work ``What is Wrong with Perplexity for Long-context Language Modeling?, ICLR 2025``.

---

> > ### Comment · Reviewer_Hihf · 2025-11-24
> >
> > Thanks for the prompt response. In my opinion, the revision is still too messy to read and I don't think my concerns towards presentation are resolved. Additionally, the revision exceeds 9 pages.

---

### Official Review · Reviewer_Ci5v · 2025-10-26

**Soundness:** 2
**Presentation:** 1
**Contribution:** 2
**Rating:** 2
**Confidence:** 3

**Summary:**

This work mainly explores a context compression approach to optimize efficiency in long-context modeling. The core idea is to compress the intermediate KV cache using various pooling strategies and combine this with sliding window attention and attention sink techniques to reduce inference complexity from O(L²) to O(L). The compression strategy is inspired by snapkv, where tokens that receive the highest attention scores from other tokens are retained by observing their attention scores.

**Strengths:**

1. This work provides comprehensive experimental data on its context compression methods.

**Weaknesses:**

1. Method Effectiveness: Many previous studies have shown that lossy context compression leads to significant information loss. As seen in Table 6, when DCA is removed, the performance of pure ILRe extrapolation on the ruler drops rapidly. This raises doubts as to whether the improvements on the ruler are mainly due to DCA.
2. Unclear Writing Structure: The paper lacks an intuitive introduction to the core ideas behind the method, instead jumping straight into the details, which makes it difficult for readers to grasp the underlying logic.
3. The Method Is Overly Simple: There are no substantial improvements in the approach—it largely relies on methods proposed in earlier work. For example, most of the context compression strategies are borrowed from snapkv.
4. Weak Baseline Comparison: A highly related work is "Sparser is Faster and Less is More: Efficient Sparse Attention for Long-Range Transformers"(https://arxiv.org/abs/2406.16747), which implements context compression in every layer and provides efficient kernels, yet this work is not discussed.

**Questions:**

See weaknesses.

---

> ### Author Response · Authors · 2025-11-14
>
> For weaknesseses:
> 1. DCA is optional in ILRe. The DCA part in the ablations is aiming to verify the compatibility with ILRe. The drop of performance without DCA is due to no length tuning for the selected models. Actually, ILRe extrapolates Qwen models up to 128K without DCA enabled.
>
> 2~3. please refer to the revision comment.
>
> 4. The work "Sparser is Faster and Less is More: Efficient Sparse Attention for Long-Range Transformers" is already discussed in the related work. But it requires fine-tuning meanwhile our approach is training free. It's not very appropriate to compare them in this work.

---

### Official Review · Reviewer_KCGQ · 2025-10-30

**Soundness:** 3
**Presentation:** 3
**Contribution:** 2
**Rating:** 2
**Confidence:** 4

**Summary:**

This paper presents a training-free method to compress input tokens and thus reduces pre-filling time and memory. The key idea of reducing memory is (1) streaming chunk prefilling up till a specific layer (2) then calculate the attention score of that layer over full context to identify the specific tokens to keep. Compared to prior work (SnapKV), the method propose to use multiple kernels (instead of a single max kernel) to aggregate the number of tokens to keep.

Experiments are conducted on two language models (Llama-3.1-8B-Instruct and Qwen-2.5-7B-Instruct) and two datasets (LongBench and RULER), showing on-par performance with using full-context.

**Strengths:**

* The paper presents a training-free method to compress tokens and thus reduce the memory and compute used for long-context scenario, which demonstrates promising performance on the dataset tested.
* The method is intuitive and presented clearly. The discovery of using one layer's attention score to perform compression is interesting.

**Weaknesses:**

1. The benchmarks are limited. While the authors have conducted evaluation on LongBench, datasets in LongBench primarily consists of relatively short context (<10K), which is relatively easy to prefill. As the paper aims to compress extremely long text, it will be good to evaluate on [Infini-Bench](https://arxiv.org/pdf/2402.13718), which consists of context of >100K.
2. The baselines are also limited. While the paper has included two KV cache compression baselines (SnapKV and StreamingLLM), I believe the following baselines are also appropriate, especially as the paper targets speed-up for pre-filling time: [DuoAttention, ICLR2025](https://arxiv.org/pdf/2410.10819) and [MMInference 1.0, NeurIPS 2024](https://arxiv.org/pdf/2407.02490). Specifically DuoAttention proposes the streaming chunked pre-filling process employed by this paper, although in a different way (for specific heads).
3. Efficiency evaluation is also limited: Figure 1 presents the TTFT for the proposed method, but it is a bit unclear what is the setting (which layer is $L_{R}% and whether it was with full KV or a chunking window? This is important to understand the performance efficiency trade-off for the method (as Table 3 shows varying downstream performance when these hyper-parameters are different). It is also good to compare efficiency with the aforementioned baseline which aims to reduce pre-filling time.

**Questions:**

* Missing experiment results: Table 1 presents the RULER results for Llama-3.1-8B-1M-Instruct, what's the RULER results for Qwen-2.5-7B? I understand the Qwen-2.5-7B has a context window of 128K, but it is ok to present results up to 128K. Also, what is the performance for RULER under different hyperparameters (layer, W, etc.)

---

> ### Author Response · Authors · 2025-11-14
>
> 1. We demonstrate the context capacity up to 1M by RULER results. LongBench is relative short but still cover most usual cases in AI applications.
> 2. DuoAttention is not easy to align compression ratios with other methods as it assigns chunking prefill in selected attention heads. For example,  we set 4K budget for other methods but DuoAttention will have no attention head left if the context is over 100K long.
>
> MInference is a good choice even though it has limited supported models. Unfortunately, it's not compatible with our hardware platform HUAWEI NPU due to its custom triton operators.
>
> 3. We add more clarification of setting in the revision.
>
> For the questions:
> Table 6 has RULER results for Qwen family. We add different $l_R$ results of RULER in Table 1.

---

> > ### Comment · Reviewer_KCGQ · 2025-11-24
> >
> > Thank you for your response. However, I don't think my concerns are addressed:
> > (1) **Limited benchmark**: While the paper conducted experiments up to 1M for RULER, it is a suite of synthetic tasks, while LongBench contains <10K inputs. I still think evaluation on Infini-Bench would be important.
> > (2)  **Limited baseline**: I think it would make sense to compare the proposed method with MInference on widely used hardware such as NVIDIA GPUs.

---

### Official Review · Reviewer_kS2A · 2025-10-31

**Soundness:** 2
**Presentation:** 1
**Contribution:** 2
**Rating:** 2
**Confidence:** 4

**Summary:**

This paper proposes ILRE, which enhances long-text performance in KV cache compression by incorporating techniques such as max pooling and multiple kernel sizes.

**Strengths:**

1. Compared with using the full context, ILRE significantly reduces TTFT.
2. It achieves performance improvements on RULER and LongBench.

**Weaknesses:**

Overall, I believe this paper does not present a fundamentally novel contribution compared with previous methods (e.g., SnapKV). The addition of techniques such as max pooling and multiple kernel sizes lacks sufficient innovation. Moreover, the writing is confusing to me.

1. Figure 1 lacks a comparison with baselines (such as StreamingLLM and SnapKV).
2. The setup in Figure 2 is unclear. What does Recall mean here? It seems that the authors placed some key information in the appendix (for example, recall in A.1), which makes the paper difficult to follow.
3. In lines 252–259, the authors introduce several optimization techniques based on SnapKV. However, they do not clearly define these optimizations or explain which key issues they address. For example, the authors repeatedly mention multiple kernel sizes—how are these kernels defined, and what does max/mean pooling of different sizes mean in this context? None of this is explained. In Section 4.3, it appears that using fewer kernels leads to worse performance, so why not increase the number of kernels?
4. The baselines used in the paper are insufficient.

**Questions:**

The results in Table 1 may be incorrect. For instance, StreamingLLM uses fixed memory consumption, so OOM (Out Of Memory) should not occur.

---

> ### Author Response · Authors · 2025-11-14
>
> For the weakness, please see the revision comment.
>
> For the question about StreamingLLM, we are using the version StreamingLLMPress from kvpress(https://github.com/NVIDIA/kvpress) which has a good community support. In its implementation, it will prefill the whole context then evict tokens. This results in no difference in the prefilling time or memory from FullContext but more accurate first generated token.

---

### Author Response · Authors · 2025-11-14
**Novel Contribution of this work**

Thanks reviewers for their time and suggestions.

Given the concerns of novelty in our work, we want to emphasize the following contributions:
1. Propose a novel approach to integrate several techniques for efficient context compression. This motivation is clearly stated in the introduction.
2. Discover the strong retrieval ability in the intermediate layers. Especially this ability happens in the early layers which can significantly reduce the computation complexity.

The new revision has the following changes:
1. rephrase ``kernel`` to ``combination``, and define one combination as a combination of max and average pooling. This happens mostly between line 87 and line 97 and in section 3.2 \& 3.3 .
2. rephrase the ``recall`` to ``retrieve`` and change ``Recall (%)`` to ``Accuracy (%)`` in the charts.
3. add more clarifications in Figure 1 and Figure 2.
4. relocate the retrieval task definition to section 3.2.
5. put more results for different $l_R$ settings in Table 1.

---

### Meta-Review · Area_Chair_SkDk · 2026-01-03

**Summary:**

The paper presents a training-free LLM’s context compression method, named Intermediate Layer Retrieval (ILRe). The key idea is stream profiling up to a specific layer, then calculate the attention score of that layer over the full context to identify which tokens to keep. This is
largely inspired by SnapKV method, with some changes (e.g., multiple kernels vs. one kernel in SnapKV).

Experiments are done on two LM (Llama-3.1-8B-Instruct, Qwen-2.5-7BInstruct) on two datasets (LongBench and RULER) showing promising performances compared to baselines (e.g., SnapKV, StreamingLLM).  The reviewers are overall positive about experimental design, such as analysis experiments.

Despite the positives, the reviewers as well as AC, is recommending the paper in its current form. Many reviewers (Hihf, Ci5v,kS2A) pointed out issues with the clarity of the paper write-up / organization. The rigor of experiments (scope of baselines/datasets evaluated) were also unsatisfactory (Reviewer KCGQ, Ci5v, Hihf). The experimental results, while positive over baseline, also still is limited compared to full context models.

**Reviewer Concerns:**

Many reviewers (Hihf, Ci5v,kS2A) pointed out issues with the clarity of the paper write-up / organization. The rigor of experiments (scope of baselines/datasets evaluated) were also unsatisfactory (Reviewer KCGQ, Ci5v, Hihf). The experimental results, while positive over baseline, also still is limited compared to full context models.

**Reviewer Scores:**

I do not think any of the reviewers would have changed the scores for this paper, given the specifically pointed out baselines/technical issues/scope of experiments were addressed during the rebuttal.

---

### Decision · Program_Chairs · 2026-01-26

Reject